# Carnitine-acylcarnitine Translocase Deficiency with c.199-10T>G Mutation in Two Filipino Neonates Detected through Parental Carrier Testing

**DOI:** 10.3390/ijns9010004

**Published:** 2023-01-11

**Authors:** Suzanne Marie G. Carmona, Mary Ann R. Abacan, Maria Melanie Liberty B. Alcausin

**Affiliations:** 1Department of Pediatrics, Philippine General Hospital, University of the Philippines Manila, Manila 1000, Philippines; 2Institute of Human Genetics, National Institutes of Health, University of the Philippines Manila, Manila 1000, Philippines

**Keywords:** carnitine acylcarnitine translocase deficiency (CACTD), fatty acid oxidation defect, c.199 10T>G mutation, SLC25A20

## Abstract

Carnitine-acylcarnitine translocase deficiency (CACTD), a fatty acid oxidation defect (FAOD), can present in the neonatal period with non-specific findings and hypoglycemia. A high index of suspicion is needed to recognize the disorder. The case is of a 24-year-old G2P2(2000) mother who sought consultation for recurrent neonatal deaths. The neonates, born two years apart, were apparently well at birth but had a fair cry and no spontaneous eye opening within the first 24 h of life and died before the 72nd hour of life. Newborn screening of both babies revealed elevated long chain acylcarnitines and hypocarnitinemia suggestive of a FAOD. However, due to their early demise, no confirmatory tests were done. Parental carrier testing was performed, revealing both parents to be heterozygous carriers of a pathogenic variant, c.199 10T>G (intronic), in the *SLC25A20* gene associated with autosomal recessive CACTD. This is the first reported case of CACTD in the Filipino population.

## 1. Introduction

Mitochondrial fatty acid oxidation is a major source of energy during fasting and excessive energy expenditure [1]. It occurs primarily in the liver and yields energy in the form of adenosine triphosphate (ATP) when acetyl–coenzyme A is degraded into carbon dioxide and water in the citric acid cycle or in the form of ketone bodies.

Carnitine-acylcarnitine translocase (CACT) is a protein encoded by the *SLC25A20* gene which has been mapped to chromosome 3p21.31 [2]. It is one of the components of the carnitine shuttle necessary for the transport of long chain acylcarnitines into the mitochondria in exchange for free carnitine [3]. 

Since CACT is present in the mitochondria of all tissues, particularly the heart, liver and skeletal muscles, CACT deficiency (CACTD) (OMIM#212138) may present with early onset hypoketotic hypoglycemia, hyperammonemia, neurologic damage, cardiomyopathy, liver dysfunction and muscle weakness [4]. It is an autosomal recessive disorder that could manifest neonatally as severe metabolic decompensation and result in rapidly progressive deterioration and death despite medical intervention [2]. 

We present the first documented case of CACTD in the Filipino population diagnosed through parental carrier testing after two consecutive neonatal deaths. This report emphasizes the importance of having a high index of suspicion in the recognition of this very rare disorder, in a well newborn with sudden deterioration, especially with a similar course in a sibling with a positive newborn screen for fatty acid oxidation disorder (FAOD).

Informed consent for the publication of medical data was secured from the couple.

## 2. Case Report

Our patient was a 24-year-old G2P2(2000) mother who consulted for recurrent early neonatal deaths. Her first pregnancy (Baby A) in 2016 was a full term (37 weeks), 2.4 kg male neonate delivered via caesarean section due to non-reassuring fetal status. The prenatal history was unremarkable. The baby was born active with good cry, however, was treated for early onset sepsis due to meconium staining. He was given unrecalled intravenous antibiotics. No IV fluids were started. On the 17th hour of life, the sleeping baby was roomed in with the mother. The family was alarmed when he continued sleeping until the 21st hour of life without waking to feed. He had no spontaneous eye opening and had fair cry. No unusual odor was noted. The family was reassured that newborns normally have a long sleeping duration, and no medical intervention was initiated. On the 33rd hour of life, the baby had sudden onset generalized cyanosis and subsequently went into cardiac arrest. The baby expired despite resuscitative measures. His death was signed out as a case of aspiration pneumonia.

The second pregnancy (Baby B) was in 2018. The pregnancy was complicated by premature contractions at 4 months age of gestation, requiring admission and tocolysis. The rest of the pregnancy was unremarkable. The baby was female, born full term (38 weeks), 2.6 kg via repeat caesarean section. She was active with good cry on delivery and latched well with the mother, thus, was roomed-in by the 4th hour of life. She was well until the 19th hour of life when she had no spontaneous eye opening with fair cry and fair suck. She was immediately admitted to the NICU and was given 10% IV dextrose infusion. Her sensorium and activity improved with no recurrence of the previous symptoms. She was discharged directly from NICU after the 48th hour of life. At home, on the 54th hour of life, she had fair suck, labored breathing and no spontaneous eye opening. This persisted until the 61st hour of life and progressed to cyanosis and sudden hypotonia. She was brought to the emergency room but was declared dead on arrival. 

The neonates were born to a healthy nonconsanguineous Filipino couple. There was no history of miscarriage, recurrent pregnancy loss and early neonatal death in other members of the family [Figure 1].

Pedigree showing two consecutive neonatal deaths born to a nonconsanguineous couple of Filipino descent. No other family members are similarly affected.

## 3. Investigations

### 3.1. Biochemical Tests

Baby A’s newborn screening was done after the 24th hour of life, and results were released after five days. Newborn screening showed elevated C16, C18, C14, C12 and C10 with low C0 (Table 1). Due to the low C0, a defect in carnitine transporter vs. Carnitine uptake deficiency was considered. 

Baby B’s newborn screening, obtained on the 30th hour of life and released after three days, showed elevated C16 and C18, with low C0. A probable Carnitine Palmitoyl Transferase 2 (CPT2) deficiency was considered (Table 2).

Due to their early and sudden demise, no confirmatory tests were done following the positive newborn screening. Upon receipt of Baby A’s positive newborn screening result, the primary health care provider did not make further referrals as the patient had already expired and his death was attributed to aspiration pneumonia. It was only after the sudden demise of Baby B and receipt of her positive newborn screening result that the family was referred to a geneticist for further evaluation.

### 3.2. Genetic Tests

Since the babies’ specimens were not available for mutational analysis, parental carrier testing was performed. Based on the babies’ clinical course of decreased activity and poor suck after a period of wellness that culminated in sudden deterioration and death, reinforced by positive newborn screen for fatty acid oxidation disorder based on abnormal long chain acylcarnitines and low carnitine, a probable FAOD was considered. Thus, parental carrier testing was performed to screen for FAODs using the Fatty Oxidation Defects panel by Invitae Company (San Francisco, CA, USA). Sequence and deletion/duplication testing of 17 common FAOD genes were performed. 

Both parents were identified to be heterozygous carriers of a pathogenic variant, c.199-10T>G (Intronic), in the *SLC25A20* gene associated with autosomal recessive CACTD. The sequence change falls in intron 2 of the gene. While it does not directly change the encoded sequence of the *SLC25A20* gene, experimental studies have shown that this intronic change causes aberrant splicing resulting to either skipping of exons 3 to 4 or of exon 3 alone. This was found to cause premature protein truncation resulting to a nonfunctional translocase enzyme [5]. The biochemical profile of CACTD, such as elevated C16, C18, C18:1 and C18:2 acylcarnitines and hypocarnitinemia, is indistinguishable from CPT2 deficiency. Thus, it is imperative to rule out mutations in the CPT2 gene. The couple had normal CPT2 genes making this condition less likely. The results of the mutation testing confirm the diagnosis of CACTD in their children.

## 4. Discussion

CACTD is a rare disorder with an incidence of 1:750,000 to 2,000,000 among Caucasians [6] but with an estimated incidence of 1:60,000 in Hong Kong due to a known recurrent mutation that may frequently present in the Southern Chinese population [7]. There are less than 100 reported cases worldwide [3,4,8]. It presents with a combination of energy depletion and endogenous toxicity affecting organs that rely on fatty oxidation for fuel; hence, its manifestations include cardiomyopathy, liver dysfunction, skeletal dysfunction, hypotonia and neurologic dysfunction [2]. 

During acute decompensation, patients with CACTD develop hypoglycemia due to hepatic glycogen depletion and impaired gluconeogenesis. Hypoketosis occurs due to deranged fatty acid transport and impaired fatty acid oxidation. Hyperammonemia arises secondary to urea cycle dysfunction due to decreased availability of N-acetylglutamate. N-acetylglutamate is diminished due to elevated propionyl CoA and decreased acetyl CoA concentrations. Creatine kinase and transaminases increase due to muscle and liver damage. Dicarboxylic aciduria is usually detected when alternative pathways are utilized in place of mitochondrial β-oxidation to oxidize excess fatty acids. Long chain acylcarnitines (C16, C18, C18:2, C18:2) are also elevated; however, this could hint to either CACTD or CPT2 deficiency [2]. In both babies, no diagnostic tests during acute decompensation were done. Newborn screening results showed abnormal acylcarnitines and hypocarnitinemia signifying a probable carnitine transporter defect or carnitine uptake deficiency on Baby A and CPT2 deficiency on Baby B. Unfortunately, the neonates expired prior to confirmatory testing.

When presented with a neonate who suddenly deteriorates after a period of wellness, the detection of hypoketotic hypoglycemia should raise the suspicion of CACTD and other FAODs [9]. Aside from glucose and ketones, plasma ammonia and lactate levels may be requested, and glucose infusion should be started immediately [10]. The diagnosis of CACTD and other FAODs can easily be overlooked especially in patients with nonspecific manifestations. Knowing that such a condition is present in our population, having a high index of suspicion is crucial in its diagnosis. Requesting the appropriate diagnostic tests is one step closer to giving the patients the necessary medical management.

Newborn screening by tandem mass spectrometry can provide an important clue in the early detection of long chain FAODs. CACTD is not included in the Philippine Expanded Newborn Screening Program (ENBS). While it is not included, its primary analytes are similar to that of CPT2 deficiency, which is part of the panel of ENBS disorders. An abnormal newborn screening for FAODs should prompt the clinician to request for acylcarnitine analysis. While the acylcarnitine profile of CACTD is similar with CPT2 deficiency, confirming the diagnosis of CACTD is done either by enzyme analysis or mutational analysis of the *SLC25A20* gene [3]. Since there was no opportunity to save specimens from the babies prior to their demise, parental carrier testing was performed, which revealed that both parents were heterozygous carriers of a c.199-10T>G mutation in the *SLC25A20* gene. A normal CPT2 gene confirms that both children were likely to have had CACTD.

At present, there are over 42 pathogenic mutations in the *SLC25A20* gene [8]. The c.199-10T>G is a splicing mutation in intron 2 of the *SLC25A20* gene and is the most common mutation among patients from China, Thailand, Japan and Vietnam [9,11,12,13,14]. Eleven CACTD patients were previously found to have this mutation in literature, and all presented with symptoms in the first three days of life. The most prevalent symptoms were sudden cardiac arrest, apnea/respiratory distress, poor feeding and lethargy [3]. Another study revealed both patients with this splicing mutation quickly deteriorated and had cardiorespiratory collapse within the first week of life despite early and aggressive treatment [3]. These reports suggest that patients with CACTD and c.199-10T>G mutation have a high mortality rate in the neonatal period.

Management goals, once a FAOD is suspected, include avoidance of fasting and IV glucose infusion with or without insulin to inhibit lipolysis and fatty acid oxidation. If hyperammonemia is detected, ammonia detoxification is warranted. A special diet low in long chain fatty acids supplemented with medium chain triglycerides (MCT) is needed to restrict long chain acylcarnitine accumulation. Carnitine therapy to address hypocarnitinemia, at this time, remains controversial [2].

The early diagnosis and treatment of CACTD is crucial. However, patients remain vulnerable every fasting period or illness to metabolic crisis that could result in neurologic sequelae and even sudden death. Long term management of patients remain to be avoidance of fasting, adherence to a low long chain fat diet with MCTs and IV glucose infusion and ammonia detoxification during acute crisis. Cardiac surveillance may also be warranted to monitor for cardiomyopathy [2]. 

CACTD is a rare disorder that was not previously reported in the Filipino population. It has seemingly nonspecific manifestations but with a high mortality in the first year of life. The genotype and enzyme residual function are important prognostic factors with the c.199-10T>G mutation predicting a severe phenotype [5]. In some cases, however, prognosis may be better in pre-symptomatic individuals identified after having an affected sibling, allowing for the individual’s early diagnosis and management [4]. In the case of our patient, the outcomes for Baby B could have been altered had the knowledge of Baby A’s course and similar clinical presentation prompted further diagnostic evaluation of the former prior to discharge. Unfortunately, Baby A’s sudden deterioration was attributed to an infectious etiology, which is by far more common, inhibiting proper referral when it was needed.

## 5. Conclusions

The couple have undergone several genetic counseling sessions following the results of their tests, which showed that they are heterozygous carriers of a pathogenic mutation in the *SLC25A20* gene. This gives them a 25% recurrence risk of CACTD in every pregnancy. Having this diagnosis could serve to improve the family’s reproductive choices through pre-implantation genetic testing or through early confirmatory testing for CACTD in the neonate and anticipatory management if warranted. Just like other disorders that are screened for, the prompt medical intervention upon diagnosis of CACTD and during acute episodes, along with good compliance to long term management, may improve a patient’s outcome.

This is the first genetically confirmed case of CACTD in the Filipino population. This case highlights the need to follow-up newborn screening results when faced with a critically ill neonate, and the benefit of proper referral to specialists even in the case of neonatal demise. Ideally, a thorough post-mortem examination should have also been done; however, this is not a common practice in the Philippine setting. It is the hope of the authors that by sharing this experience, we can broaden the knowledge regarding rare disorders, such as CACTD. Having a high index of suspicion for CACTD and other FAODs when similar manifestations are encountered should improve the chances for prompt diagnosis and management, knowing that this condition is present in the Filipino population.

## Figures and Tables

**Figure 1 IJNS-09-00004-f001:**
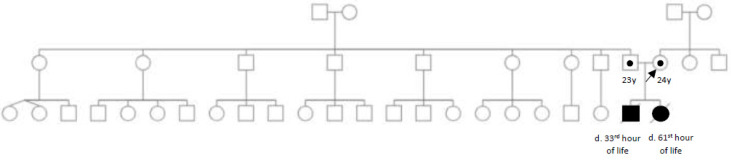
Pedigree.

**Table 1 IJNS-09-00004-t001:** Newborn Screening Result of Baby A.

µmol/L	At 24 h of Life
C0 (≥10.85)	9.45
C2 (≥3.35)	12.67
C14 (≤0.58)	1.74
C16 (≤6.22)	19.98
C18 (≤1.87)	3.75
C10 (≤0.4)	0.98
C12 (≤0.4)	1.27

**Table 2 IJNS-09-00004-t002:** Newborn Screening Result of Baby B.

(µmol/L)	At 30 h of Life
C16 (<7)	15.06
C18 (<2.3)	2.75
C18:2 (<0.98)	0.20
C16:1 (<0.55)	1.21
CPT2 (<27.85)	0.29
C18/C3 (≤2.15)	4.63
C18:1/C8 (≤66)	24.89
C0/(C16+C18) (>1.34)	0.37
C18:1 (<3.04)	3.49
C0 (≥100)	6.46

## Data Availability

No new data were created in this case report. All relevant medical information were mentioned in the manuscript.

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
