# Peer review of "Carnitine-acylcarnitine Translocase Deficiency with c.199-10T>G Mutation in Two Filipino Neonates Detected through Parental Carrier Testing"

_2409-515X, 2023, doi:10.3390/ijns9010004_

Round 1

Reviewer 1 Report

Very well written and interesting manuscript. 

Minor comments/questions

- Transcript ID should be provided somewhere; I assume transcript NM_000387.6.

- Not necessary, but full NBS/tandem mass spectrometry results, could be of interest to the community; even if provided as supplemental table etc.

Author Response

Good day. The following are my responses to the reviewer's comments:

  1. Manuscript ID is the file name used for the document uploaded.
  2. I will include the newborn screening results of the babies as Tables to be incorporated in the Biochemical Tests done.

Reviewer 2 Report

In the submitted case series, the authors report recurrent neonatal deaths suspected to be caused by carnitine-acylcarnitine translocase (CACT) deficiency in a single family. The authors state this is the first case of CACT deficiency documented in patients who are Filipino.  The case report also demonstrates the importance of biochemical testing (in this case newborn screening). Most notably, the case report emphasizes the importance of follow up abnormal newborn screen results even in the cases of neonatal deaths.  As a result, this case report will be of interest to the readers of the International Journal of Neonatal Screening.

Major Comment

The case report leaves the reader (or at least this reviewer) to wonder why the initial abnormal newborn screen was never followed up.  The authors stated it was due to the neonatal deaths.  If this is common practice or policy, it would be helpful to discuss.

Furthermore, this is a great opportunity for the authors to question that practice – policy.  It is possible that follow up of the first newborn screen (even with parental testing as described here) could have prevented to the second neonatal death. 

Minor Comments:

Case report:

The second child was discharged home after a brief admission to the neonatal intensive care unit and receiving IV dextrose. The child then re-presented shortly after discharge. It is assumed, at least by the reviewer, that the child was not feeding well (or at all) after discharge. But this would be helpful to document.

Similar cases

·         Of note, a few similar cases have been published of neonatal death with long chain fatty acid oxidations disorders where the only option was parental sequencing to ‘confirm’ the screening result (my memory is that they are all VLCADD).  These cases would support the approach by the authors.

Author Response

Good day. The following are our responses to the reviewer's comments:

Major Comment

The case report leaves the reader (or at least this reviewer) to wonder why the initial abnormal newborn screen was never followed up.  The authors stated it was due to the neonatal deaths.  If this is common practice or policy, it would be helpful to discuss.

---Upon receipt of Baby A’s positive newborn screening result, the primary health care provider did not make further referrals as the patient has already expired and her death was attributed to aspiration pneumonia. It was only after the sudden demise of Baby B and receipt of his positive newborn screening result that the family was referred to a geneticist for further evaluation. Since the sudden deterioration of the first baby was attributed to infectious causes, this inhibited proper referral when it was necessary.

Furthermore, this is a great opportunity for the authors to question that practice – policy.  It is possible that follow up of the first newborn screen (even with parental testing as described here) could have prevented to the second neonatal death. 

---The prognosis of patients with CACTD may be better in pre-symptomatic individuals diagnosed after having an affected sibling. Indeed, this could have been the case for Baby B had Baby A's history and newborn screen been adequately reviewed. We will incorporate in the conclusions that this highlights the need to follow-up newborn screening results when managing critically ill neonates, and that there is still benefit in referring to specialists even in the case of neonatal demise. Ideally, post-mortem examination should have been done, however, this is not a common practice in the Philippine setting. 

Minor Comments:

Case report:

The second child was discharged home after a brief admission to the neonatal intensive care unit and receiving IV dextrose. The child then re-presented shortly after discharge. It is assumed, at least by the reviewer, that the child was not feeding well (or at all) after discharge. But this would be helpful to document.

--Yes, the baby was not feeding well, had labored breathing and did not have spontaneous eye opening. We will include that in the manuscript.